# Band-Stop Frequency-Selective Surface (FSS) with Elliptic Response Designed by the Extracted Pole Technique

**DOI:** 10.3390/s24144452

**Published:** 2024-07-10

**Authors:** José R. Montejo-Garai, Juan E. Page, Gerardo Perez-Palomino, Robert Guirado

**Affiliations:** Group of Applied Electromagnetics (GEA), Information Processing and Telecommunications Center, Universidad Politécnica de Madrid, 28040 Madrid, Spain

**Keywords:** band-stop, extracted pole, frequency-selective surface (FSS), spatial filter, transmission zero, unit cell

## Abstract

This paper describes and validates an advanced synthesis design process of Frequency-Selective Surfaces (FSSs) with elliptic band-stop responses. A systematic procedure based on the Generalized Chebyshev Function and the extracted pole technique enables control of the position of the transmission zeros and the attenuation level to obtain an equiripple rejection response. A systematic process is followed to obtain the lumped LC values of the resonator circuits extracted as poles and the impedance inverters. Then, equivalent dipoles and transmission lines are obtained to carry out the electromagnetic design at normal incidence for a linearly polarized field. The impact of the higher-order modes of the periodic structure on the electrical response of the FSS, which can be relevant due to the stringent selected specifications, has been also analyzed. A fourth-order band-stop filter with a 3 GHz bandwidth centered at 30 GHz and its attenuation at 50 dB has been designed considering three different implementations: two filters using a vacuum as a transmission line with different connection lengths and a third one using a dielectric substrate to enable its manufacturing. In order to verify the design procedure using experimental results, the third filter with printed dipoles in the dielectric substrate has been manufactured and measured, thus validating the developed process.

## 1. Introduction

Frequency-Selective Surfaces (FSSs) have been investigated for decades [1] because of their potential to construct radomes, antenna reflectors, spatial filters, absorbents, electromagnetic shelters, artificial magnetic conductors, and electromagnetic band-gap materials, from RF to millimeter waves [2,3,4].

FSSs are implemented by means of arrays of different geometries such as dipoles, circular rings, multiple kinds of crosses, patches, or even random patterns, thus supported by a dielectric substrate. Depending on the chosen geometry, the frequency response may be low-pass, high-pass, band-stop, or band-pass [5].

Because of their frequency-selective behavior, FSSs are commonly used in communication systems to efficiently control the transmission and reflection of electromagnetic waves in free space conditions. In addition, the FSSs as periodic structures excited by a plane wave at a certain angle of incidence allow for band-pass or band-stop responses with the bandwidth and attenuation as design parameters [6]. In this context, FSSs with band-stop performance are a key element that is increasingly integrated in wireless communication systems to separate contiguous local area networks or channels in frequency [7,8].

As it is known, FSSs based on a single layer lack high selectivity and bandwidth [9]. As a consequence, the connection of several layers, including dielectric spacers, is the most common way to increase the bandwidth and, at the same time, improve the attenuation response [10].

There are different approaches in the technical literature to achieve band-stop FSSs, but most of them use direct brute-force optimization of structures following heuristic arguments, or they employ trial and error processes [7,11,12,13,14,15,16]. On the other hand, an equivalent circuit model has been proposed for a larger fractional bandwidth response [17]. Different 3D structures with multiple transmission zeros or multiband responses are shown in [18,19,20].

However, in this scenario, a systematic synthesis procedure to design a band-stop FSS with total control of the bandwidth and the attenuation level is missing. In this work, the powerful extracted pole technique has been applied, for the first time to the authors’ knowledge, we have designed an FSS with band-stop behavior. Previously implemented in waveguide technology [21], its relevant advantages can be transferred now to open structures with periodicity conditions. In this sense, the total control of the position of the transmission zeros results in a equiripple attenuation bandwidth specified by the designer, who controls *a priori* the response by means of an analytical process that is well established in circuit theory [22].

The paper is organized as follows. Section 2 presents the theoretical synthesis of the band-stop filter using the Generalized Chebyshev Function and the extracted pole technique to obtain an elliptic response, i.e, with controlled transmission zeros as those obtained using the Jacobian elliptic functions [22]. Using this procedure, three different implementations of an FSS band-stop filter at the Ka band have been proposed, and one of them has been manufactured and measured for experimental validation purposes. In Section 3, the dipoles implementing the LC resonators previously obtained are calculated using the unit cell with the suitable period to avoid the disturbance of higher-order modes. Then, in Section 4, the full-wave design and optimization of the structure with a dielectric substrate is carried out. Section 5 shows the experimental results, including an evaluation of the technology and measurements. In addition, the results of the developed design method are confronted with comparable designs in the technical literature. Finally, the contributions of the proposed work are summarized in Section 6.

## 2. Theoretical Synthesis Procedure

A band-stop filter response is necessary when a specific band of frequencies must be attenuated. Although this task can also be carried out by band-pass filters, when interfering frequencies are particularly strong, special actions have to be considered to suppress them. In these case, the use of band-stop filters with high attenuation of unwanted frequencies is imperative.

Classical responses like Butterworth or Chebyshev can be implemented using the well-developed insertion loss synthesis procedure. The circuit values for the maximally flat attenuation or equirriple response are obtained from tables or recursively computed [5]. After the reactive transformation from the normalized low-pass prototype to the band-stop filter is carried out, the frequency response is obtained.

However, the above classical responses do not allow for the specific control of the attenuation over the desired bandwidth. In order to achieve this goal, a technique based on the extracted pole technique is presented in this work, thus introducing controlled transmission zeros to obtain an equirriple attenuation level in the stopband [22]. This synthesis technique, which has been successfully accomplished in waveguide technology [21], is now extended to the spatial configuration of an FSS. Basically, the concept relies on the interchange of responses between the transmission and reflection coefficients $S_{21}\left(s\right)$ and $S_{11}\left(s\right)$, respectively. Therefore, the return loss response becomes the insertion loss and vice versa. In this way, the synthesis process begins by finding the order of the filter *N*, which is the same value as the number of transmission zeros that fulfills the specification concerning the attenuation level. The position of the transmission zeros in the imaginary axis of the complex plane is determined by the Generalized Chebyshev Function [22].

The synthesis process extracts the transmission zeros at the initial position of the reflection zeros, thus obtaining the value of the residues and the phase shifters. In addition, the sequence of the extraction provides different residue values, although they provide the same response. Since the residue value specifies the impedance level of every resonator after the frequency transformation, it is worth analyzing different sequences of extraction in order to find the most suitable to provide the appropriate values for the physical implementation of the unit cell (dipole, cross, etc.) of the FSS.

After extracting the poles with their respective residues and the phase lengths, the synthesis process ends with the extraction of a parallel inverter. Table 1 shows the specification of the band-stop filter, with an equirriple attenuation level of 50 (dB) in a 10% relative bandwidth centered at 30 (GHz). These demanding specifications are a litmus test for an FSS.

Figure 1 shows the topology of the normalized prototype and the circuit elements after systematic extraction that verify the specifications collected in Table 1. Figure 2 shows the response of the normalized band-stop filter (S parameters).

Once the normalized band-stop prototype is synthesized, the next step is to accomplish the transformation to the center frequency $f_{o}$ and bandwidth BW specified in Table 1. In this transformation, the first and last phase shifters needed to extract the first and fourth poles can be eliminated, since the magnitude response remains unchanged. In addition, every inductor in the normalized prototype is converted to a series LC resonant circuit with values given by Equations (Equation 1) and (Equation 2). In this process, the four resonance frequencies are modified by the residue of every pole [23] to accomplish the desired transmission zero.
(1)$$L_{k}=\frac{1}{\omega_{o}b_{k}}\left(\alpha -\frac{\omega_{k}}{2}\right)\;\;\;\;\;\alpha =\frac{\omega_{o}}{BW}\;\;\;\;\;\omega_{o}=2\pi f_{o}$$
(2)$$C_{k}=\frac{1}{\frac{\omega_{o}}{b_{k}}\left(\alpha +\frac{\omega_{k}}{2}\right)}\;\;\;\;\;\alpha =\frac{\omega_{o}}{BW}\;\;\;\;\;\omega_{o}=2\pi f_{o}\;$$

Figure 3 shows the topology of the extracted pole *k*, which is composed by the inductor ($1/b_{k}$) and the invariant reactance ($-j\omega_{k}/b_{k}$) of the normalized filter and the series LC resonator after the transformation composed by the inductor ($L_{k}$) and the capacitor ($C_{k}$) obtained by means of Equations (Equation 1) and (Equation 2). Figure 4 shows the topology of the band-stop filter after frequency transformation. It is composed of the cascade connection of series LC resonators in parallel arrangement and impedance inverters. As can be observed, the input–output impedance level is scaled to that of the vacuum level η=120π(Ω). The circuit element values and the frequencies of the four transmission zeros are shown in the figure caption. The response of the band-stop filter is shown in Figure 5. The solid lines correspond to the case where the impedance inverters are ideal, i.e., frequency invariant. The dashed lines correspond to the case where the impedance inverters are implemented as transmission lines with λ/4 electrical length. As can be seen, the main difference lies in the return loss level, but the attenuation response remains unchanged.

Once the theoretical synthesis is accomplished, the next task is to implement the four resonators, the dipoles in this design, which are able to be achieved in the FSS and have the same response as the four lumped LC resonators.

## 3. Unit Cell of the FSS and Equivalent Circuits

With the intention of achieving the band-stop response by means of the FSS, distributed resonators equivalent to those lumped LC resonators shown in Figure 4 must be obtained. In the case of the inverters, their equivalent elements are transmission lines of electrical length nλ/4, with *n* being an odd number. Figure 6 shows the transmission response of the four LC lumped resonators, and the values of the elements are shown in the caption of Figure 4.

These partial responses are crucial to finding the physical dimensions of the four dipoles and the period of the unit cell. In order to obtain these dimensions from the equivalent circuit, a systematic procedure detailed in [24,25] is accomplished. In this filter, the design process requires that the two fundamental modes of the periodic structure (homogeneous plane waves propagating at a certain angle of incidence) are not coupled to each other. At normal incidence, the two fundamental modes of the TEM, which are plane waves with vertical V and horizontal H polarization, are decoupled when the metallic scatters exhibit double symmetry with respect to the period. Thus, each mode can be independently represented by a two-port network. That is the reason for choosing symmetric and centered dipoles as scatters. Note that although the design is performed for *V* polarization, the same process can be used to obtain the same electrical filtering response for the *H* polarization.

The LC lumped resonators are implemented by means of dipoles, as shown in the inset of Figure 7. Every dipole is characterized by two parameters, i.e., length and width to fit simultaneously the resonant frequency and the impedance level (zero thickness is considered). In addition, the period of the unit cell is chosen to control the cutoff frequencies of the higher-order modes.

As a validation proof, Figure 7 shows an excellent comparison between the circuit response of one of the four LC resonators and the full-wave response of its equivalent dipole simulated in a square parallel plate waveguide with boundary conditions corresponding to normal incidence for a vertical polarization. In order to dramatically reduce the computational effort, only one quarter of the unit cell is considered thanks to the two symmetry planes, the vertical with the magnetic wall condition, and the horizontal with the electric wall condition.

Following this procedure, the dimensions of the four dipoles are obtained. Therefore, the last step is the connection of the dipoles by means of transmission lines, which is implemented in free space periodic structures by means of dielectric spacers. Two different designs using electrical lengths λ/4 and 3λ/4 are accomplished with the same four dipoles.

Figure 8 shows the full-wave response of the band-stop FSS with a λ/4 length connection between dipoles (S parameters), including all dimensions, as well as the layout of the unit cell. As can be observed, the bandwidth is drastically reduced, since two transmission zeros have disappeared. The reason lies in the disturbance due to the higher-order modes. In the simulation, three modes have been taken into account in the input–output ports. In the excitation mode, the TEM mode that represents the plane wave, and the first two higher-order modes, TE and TM modes with the same cutoff frequency of 40 (GHz), result in the period or its equivalent, and the dimension of the square wavequide is 7.5 (mm). In this context, the $s_{1(2)1(1)}$ and $s_{1(3)1(1)}$ responses represent the reflection coefficients in the input port of the two higher modes. Their values are lower than −30 (dB) in the band of interest, but since we are dealing with a band-stop FSS structure, their level is irrelevant. On the contrary, the $s_{2(2)1(1)}$ and $s_{2(3)1(1)}$ responses represent the transmission coefficients in the output port of the two higher modes. As can be observed, their values are higher than −50 (dB) in the upper part of the band of interest just where the two transmission zeros have disappeared. Thus, the higher-order modes disturbance modifies the expected rejection in the band-stop filter.

The dimension of the square waveguide (the period of the FSS) is a design parameter to obtain the dimensions of the dipoles, but at the same time, it determines the cutoff frequencies of the higher-order modes. Thus, there are two ways to minimize the higher-order interaction.

The first one is to reduce the period of the unit cell in order to increase the cutoff frequencies of the higher-order modes, thus reducing the disturbance. Following this idea, a redesign of the filter is carried out, where the period decreases from 7.5 (mm) to 6.75 (mm). The new full-wave response and dimensions are shown in Figure 9. As can be observed, the level of the transmission coefficients $s_{2(2)1(1)}$ and $s_{2(3)1(1)}$ is lower than −54 (dB) in the band of interest.

The second one is to extend the connection between dipoles from λ/4 to 3λ/4. Figure 10 shows the full-wave response of a band-stop FSS with distances between layers close to 3λ/4 in length connection between dipoles (S parameters) and the layout of the unit cell (the dimensions are the same as in the previous cases). In this design, the level of the transmission coefficients $s_{2(2)1(1)}$ and $s_{2(3)1(1)}$ is lower than −60 (dB) in the band of interest. As a consequence, with this attenuation level, the four transmission zeros are in the response, since the higher-order mode’s effect is negligible.

Accordingly, a systematic procedure to design a band-stop FSS has been exposed in detail. The simulation of the unit cell by means of the equivalent waveguide allows us to control the higher mode interaction. In the next section, a band-stop filter implemented with dielectric substrate as support for the four layers of dipoles will be designed, thus taking as the starting point the results obtained in this section.

## 4. Full-Wave Design and Optimization

Once the design of the two band-stop filters with a vacuum as thetransmission line connecting the layers is carried out, it is necessary to introduce the dielectric substrate with the printed dipoles for manufacturing. Therefore, a redesign of the dipole dimensions and the period of the unit cell is imperative to take into account the thickness and permittivity of the dielectric substrate. Figure 11 shows the layout of the band-stop filter composed of four layers with printed dipoles on a dielectric substrate and three layers of Rohacell [26] (a foam with permittivity close to unity), to ensure a compact structure.

A systematic full-wave procedure is followed to adjust the response corresponding to the first design with the distance between dipoles λ/4 (Figure 9) to the new physical structure using a dielectric substrate with parameters: a thickness 0.762 (mm) and a permittivity 3.2. At this point, the circuit model is no longer used, since the higher-order modes are not enough attenuated to consider a mono mode behavior. However, the initial circuit design is an excellent starting point for the optimization.

In the optimization process, the vacuum permittivity is increased gradually from 1 to 3.2, and the thickness of the vacuum transmission line is decreased. In addition, the period of the unit cell must be considerably reduced to minimize the higher-order modes’ interactions, thus eliminating their harmful effect, as was detailed in the previous section.

By means of an optimization scheme with initial dimensions in the caption of Figure 9, the new values of the lengths and widths of the dipoles are found. The Rohacell thickness is fixed to 1 (mm), because it is a standard dimension available from the supplier. Figure 12 shows the full-wave simulation and the final dimensions of the dipoles in the optimized structure ready for manufacturing and measurement. In order to take into account the accuracy of the etching process [27] of the dipoles in the four layers, a sensitivity analysis considering a normal distribution with ±50 μm of variation in the lengths and widths has been carried out. Figure 13 shows the range of variation of the rejection level by means of the $|s_{21}|$ parameter. In addition, the influence of the thickness of the three Rohacell layers has been studied considering a normal distribution variation of ±80 μm. Figure 14 shows that the range of variation of the rejection level, by means of the $|s_{21}|$ parameter, has little relevance.

On the other hand, Figure 15 shows the simulation of the rejection response, i.e., the $|s_{21}|$ parameter considering the oblique incidence for four different angles and TE excitations: (θ=10°, ϕ=0°), (θ=20°, ϕ=0°), (θ=30°, ϕ=0°), and (θ=45°, ϕ=0°). As it can be observed, there is a progressive degradation in the rejection level decreasing as the θ angle increases. In addition, the rejected band shifts to lower frequencies. Figure 16 shows the same simulation but considering oblique incidence for four different angles and TM excitations: (θ=10°, ϕ=90°), (θ=20°, ϕ=90°), (θ=30°, ϕ=90°), and (θ=45°, ϕ=90°). In this case, the rejected band shifts to higher frequencies. Although the geometry of the cell is symmetrical with respect to the period, the TE (ϕ=0°) and TM (ϕ=90°) excitations, which exhibit an E-field polarized in the direction of the dipoles, are different. That is the reason for the differences between the results in Figure 15 and Figure 16.

However, it must be made clear that the design has been accomplished for normal incidence. In addition, note that a design considering oblique incidence requires of additional effort to control *a priori* the specifications by means of a synthesis procedure. For instance, in the best case, i.e., symmetrical metallizations and angles of incidence in the main planes ((ϕ=0°) or ϕ=90°)), a two-port circuit to electrically describe the TE wave (or the TM) would require transmission lines whose characteristic impedance is the TE (or TM) wave impedance (different from the terminal impedances, ($\eta_{o}=120\pi$)). Therefore, the synthesis process becomes more elaborate and specific for each angle of incidence. In the worst case (nonsymmetrical metallizations or angles outside the main planes), the equivalent circuits must be octopoles [25], and the design of the FSS would require us to consider synthesis methods of four-port networks (octopoles).

## 5. Experimental Results

In order to experimentally verify the design procedure exposed in the previous sections, the band-stop filter was painstakingly measured. Figure 17 shows the measurement bench composed of the VNA and the horn antennas with lenses to obtain a locally plane wave impinging normally over the FSS, which allows its accurate characterization compared to an impinging spherical wave. The manufactured FSS is shown in the inset.

The first step was to characterize every layer separately for the purpose of controlling the suitable configuration of the measuring setup. Figure 18 shows the comparison between the full-wave simulation and the measurement of the transmission coefficient $|s_{21}|$ for the four layers. As can be observed, the agreement is very good, and every notch is located at its resonant frequency, although with slight differences in the levels. The second step was to characterize the behavior of two layers, with the interaction between two notches. Figure 19 shows the comparison between the full-wave simulation and the measurement of the transmission coefficient $|s_{21}|$ for three groups of two stacked layers: 1&2, 2&3, and 3&4. In this case, some differences can be noted—most likely related to positional inaccuracies between the alignment of the stacked layers and the aforementioned resonance level mismatch.

Figure 20 shows the comparison between the full-wave simulation and the measurement of the transmission coefficient $|s_{21}|$ and the reflection coefficient $|s_{11}|$ of the band-stop filter. In order to understand this result, it is worth noting that the two-port *TRL* calibration process used results in a value of 42 (dB) for the isolation in the measurement bench. As can be observed, this level is specifically the value measured for the attenuation in the stopband, and the slope corresponds to a fourth-order response. Thus, there is a good agreement between the electromagnetic simulation and the theoretical response, which validates the developed designing procedure. In addition, the discrepancy between the measurement and the simulation can be explained by the isolation level and the assembly of the layers of the FSS. Note that the sensitivity analyzed in Figure 13 and Figure 14 assumed that the distortion introduced by the tolerances was homogeneous, thus maintaining the planar structure. However, the device bends and air subvolumes are typically generated, which could not be considered in the simulation.

Finally, Figure 21 and Figure 22 show the comparison between the measurement and the simulation at oblique incidence and TE and TM polarizations for several angles. A coherent behavior considering the theoretical results in Figure 15 and Figure 16 can be accepted. However, it is of utmost importance to highlight that this FSS was designed only at normal incidence, thus being necessary other synthesis methods and a much more complex equivalent circuit to perform a suitable design at a desired angle of incidence [25].

A comparison of different 2D-FSSs implemented in PCB technology with the band-stop response at normal incidence is shown in Table 2 (in the case of Reference [9], there are only theoretical results without measurements). It compares, among other parameters, the design process, with an emphasis on the *a priori* capability to control the desired specifications, i.e., the filter order, the bandwidth, the attenuation level and the frequency of the transmission zeros. This is the main contribution of this work. Also, note that the synthesis procedure makes it possible to know in advance the number of layers required to fulfill the required specifications, which is contrary to brute-force optimizations or trial-and-error methods (heuristic), for which the design process can be unproductive if the number of layers is arbitrarily selected.

## 6. Conclusions

An advanced design process for the synthesis of an FSS with an equirriple band-stop response has been exposed in detail. The Generalized Chebyshev Function is the starting point to apply the extracted pole technique to control the position of the transmission zeros. Therefore, the total control of the position of the transmission zeros results in an equiripple attenuation bandwidth specified by the designer.

The main contribution of this work lies in the fact that the analytically synthesized circuit was used as a predesign, thus avoiding the heuristic, brute-force, and trial-and-error methods that are so widespread in the technical literature.

The comparison between the electromagnetic simulation and the measurements validates the design process of the fourth-order elliptic band-stop filter in the Ka band. The exposed procedure was controlled throughout, from the synthesis of the lumped circuit to its implementation by dipoles, thus allowing for the achievement of the required specifications. 

## Figures and Tables

**Figure 1 sensors-24-04452-f001:**
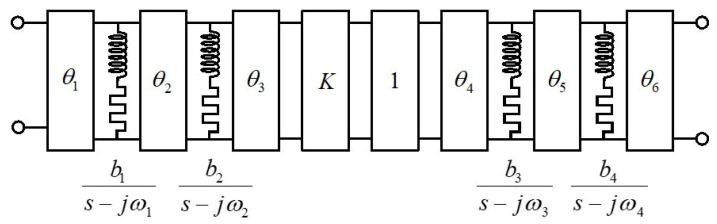
Normalized band-stop filter topology and circuit values. Order N=4. Transmission zeros: $\omega_{1}=-0.9239$, $\omega_{2}=-0.3827$, $\omega_{3}=0.3827$, $\omega_{4}=0.9239$. Residues: $b_{1}=b_{4}=1.8428$, $b_{2}=b_{3}=4.4489$. Phase shifters: $\theta_{1}=62.3560{}^{\circ}$, $\theta_{2}=-90{}^{\circ}$, $\theta_{3}=-62.3560{}^{\circ}$, $\theta_{4}=-27.6440{}^{\circ}$, $\theta_{5}=90{}^{\circ}$, $\theta_{6}=-62.3560{}^{\circ}$. Inverter: K=1.

**Figure 2 sensors-24-04452-f002:**
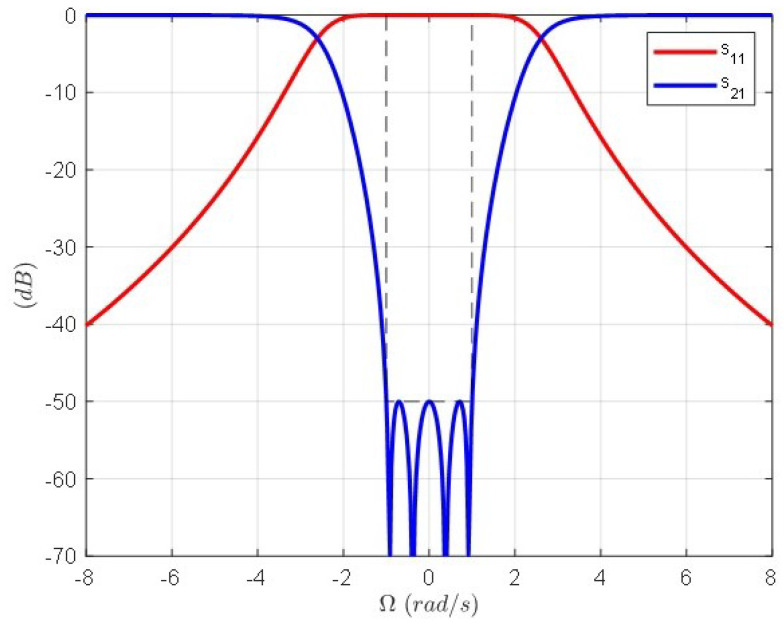
Normalized band-stop filter response of the circuit shown in Figure 1.

**Figure 3 sensors-24-04452-f003:**
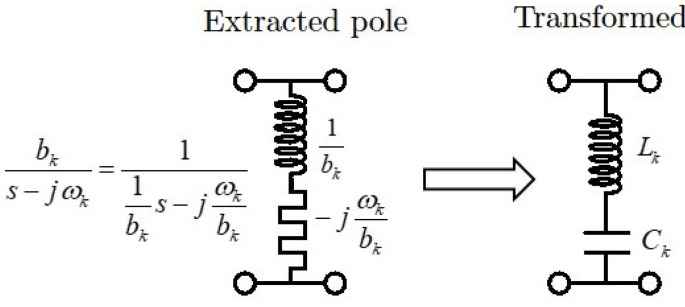
Extracted pole equivalent circuit composed by the inductor ($1/b_{k}$) and the invariant reactance ($-j\omega_{k}/b_{k}$), as well as its transformed series LC resonator composed by the inductor ($L_{k}$) and the capacitor ($C_{k}$) obtained by means of Equations (Equation 1) and (Equation 2).

**Figure 4 sensors-24-04452-f004:**
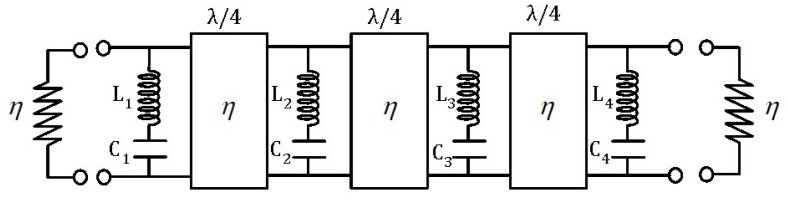
Band-stop filter topology and circuit values. Resonator 1: $L_{1}$ = 11.35 (nH), $C_{1}$ = 2.72 (fF), $f_{tz1}$ = 28.64 (GHz). Resonator 2: $L_{2}$ = 4.58 (nH), $C_{2}$ = 6.38 (fF), $f_{tz2}$ = 29.43 (GHz). Resonator 3: $L_{3}$ = 4.41 (nH), $C_{3}$ = 6.14 (fF), $f_{tz3}$ = 30.58 (GHz). Resonator 4: $L_{4}$ = 10.35 (nH), $C_{4}$ = 2.48 (fF), $f_{tz4}$ = 31.42 (GHz).

**Figure 5 sensors-24-04452-f005:**
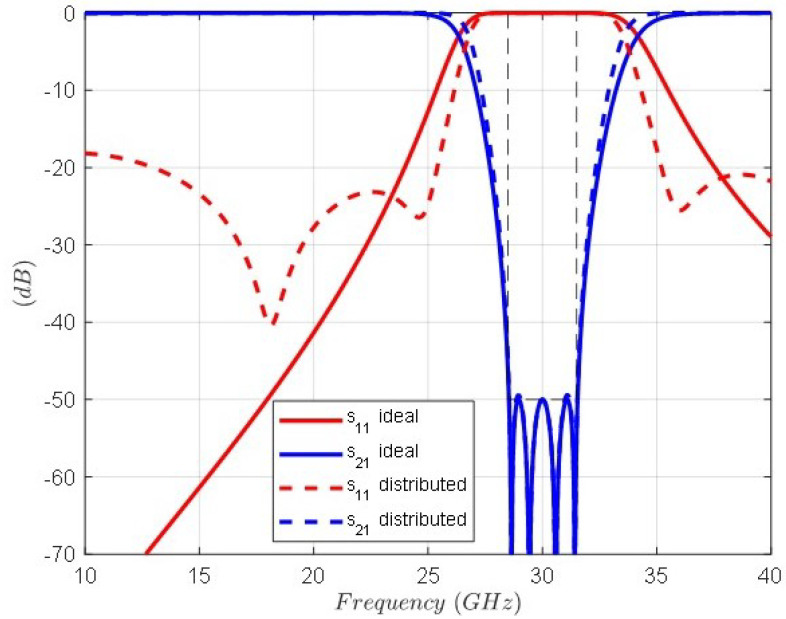
Band-stop filter response: solid lines considering ideal impedance inverters (frequency invariant) and dashed lines considering impedance inverters implemented as transmission lines with λ/4 electrical length.

**Figure 6 sensors-24-04452-f006:**
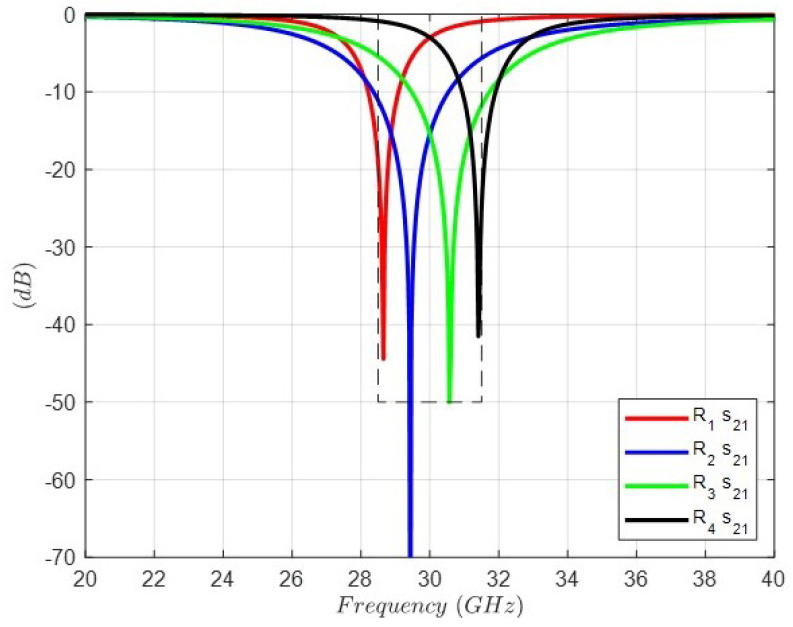
Transmission response of the four LC lumped resonators. The values of the circuit elements are shown in the caption of Figure 4.

**Figure 7 sensors-24-04452-f007:**
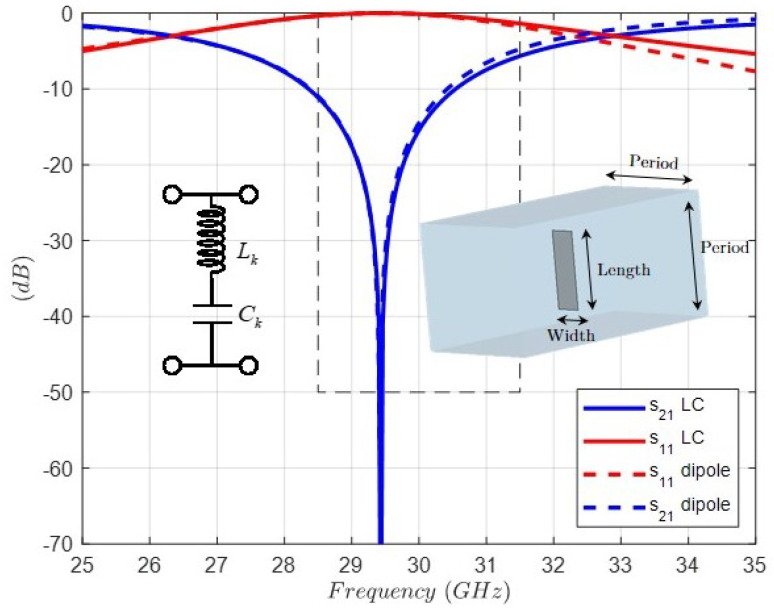
Comparison between the circuit response of one of the four LC resonators and the full-wave response of its equivalent dipole inside the unit cell. Resonator LC; $L_{k}$ = 4.58 (nH); $C_{k}$ = 6.38 (fF). Dipole dimensions: Length = 4.64 (mm), Width = 1.54 (mm), Period = 7.5 (mm).

**Figure 8 sensors-24-04452-f008:**
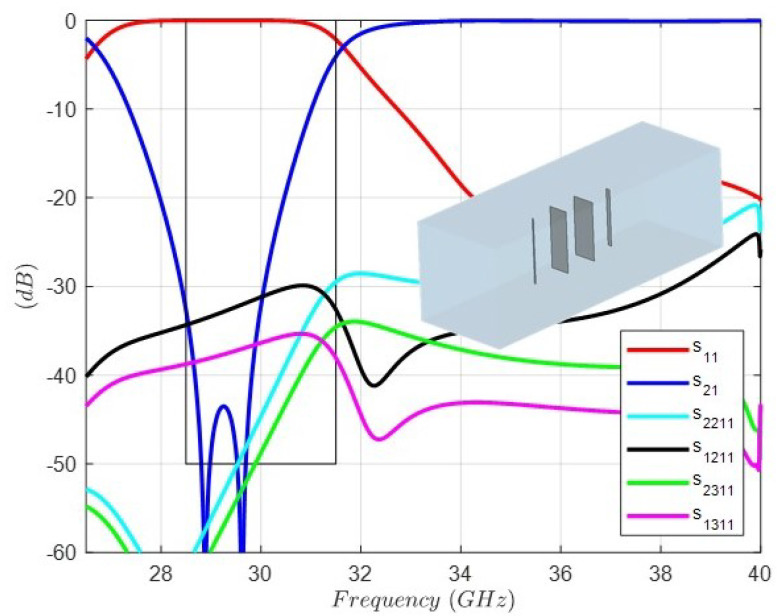
Full-wave response of the FSS unit cell topology and circuit values for the band-stop response with distance between dipoles λ/4 = 2.5 (mm). Dipole 1: $Length_{1}$ = 4.93 (mm), $Width_{1}$ = 0.125 (mm). Dipole 2: $Length_{2}$ = 4.641 (mm), $Width_{2}$ = 1.538 (mm). Dipole 3: $Length_{3}$ = 4.428 (mm), $Width_{3}$ = 1.783 (mm). Dipole 4: $Length_{4}$ = 4.27 (mm), $Width_{4}$ = 0.333 (mm). Period: 7.5 (mm).

**Figure 9 sensors-24-04452-f009:**
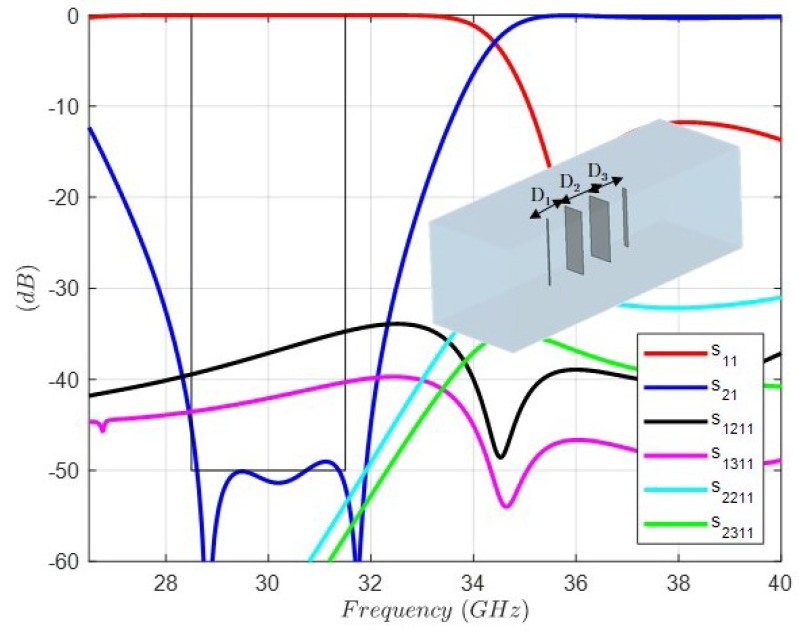
Full-wave response of the FSS unit cell topology and circuit values for the band-stop response with distances between layers close to λ/4.$D_{1}$ = 2.630 (mm), $D_{2}$ = 2.611 (mm), $D_{3}$ = 2.782 (mm). Dipole 1: $Length_{1}$ = 5.031 (mm), $Width_{1}$ = 0.152 (mm). Dipole 2: $Length_{2}$ = 4.782 (mm), $Width_{2}$ = 1.591 (mm). Dipole 3: $Length_{3}$ = 4.493 (mm), $Width_{3}$ = 1.993 (mm). Dipole 4: $Length_{4}$ = 4.025 (mm), $Width_{4}$ = 0.349 (mm). Period: 6.75 (mm).

**Figure 10 sensors-24-04452-f010:**
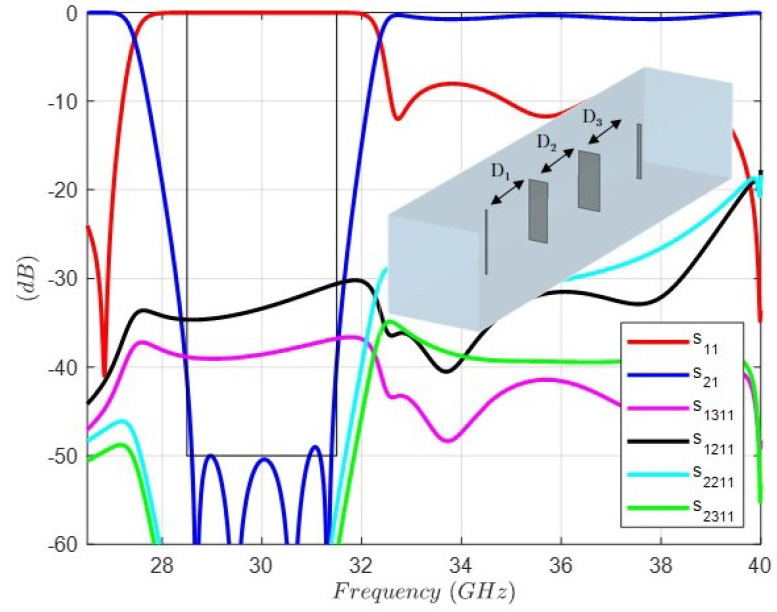
Full-wave response of the FSS unit cell topology and circuit values for the band-stop response with distances between layers close to 3λ/4.$D_{1}$ = 7.348 (mm), $D_{2}$ = 7.483 (mm), $D_{3}$ = 7.954 (mm). Dipole 1: $Length_{1}$ = 4.837 (mm), $Width_{1}$ = 0.128 (mm). Dipole 2: $Length_{2}$ = 4.609 (mm), $Width_{2}=1.574$ (mm). Dipole 3: $Length_{3}$ = 4.393 (mm), $Width_{3}$ = 1.829 (mm). Dipole 4: $Length_{4}$ = 4.323 (mm), $Width_{4}$ = 0.333 (mm). Period: 7.5 (mm).

**Figure 11 sensors-24-04452-f011:**
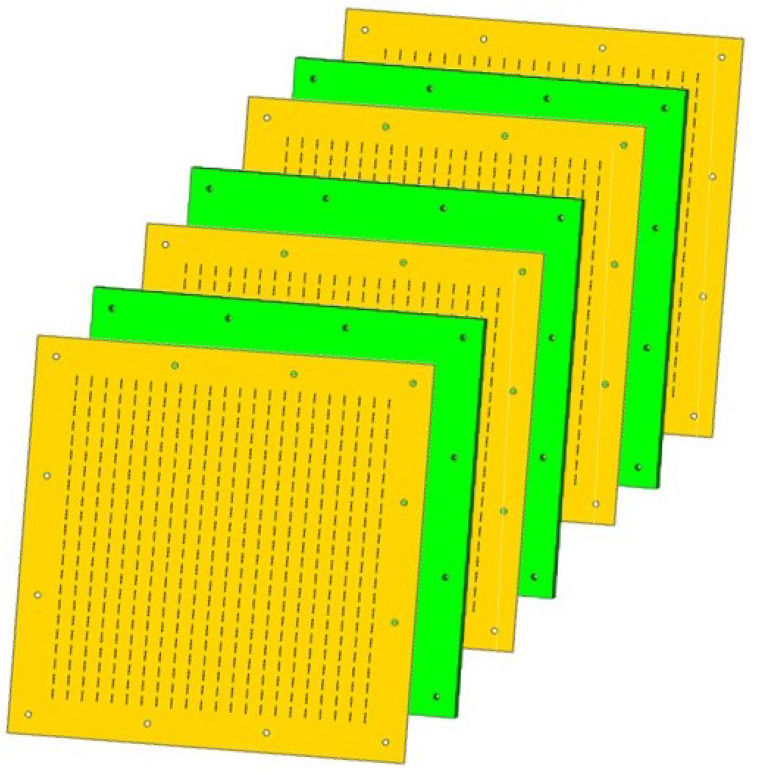
Layout of the band-stop filter composed of four layers of printed dipoles on a dielectric substrate and three layers of Rohacell.

**Figure 12 sensors-24-04452-f012:**
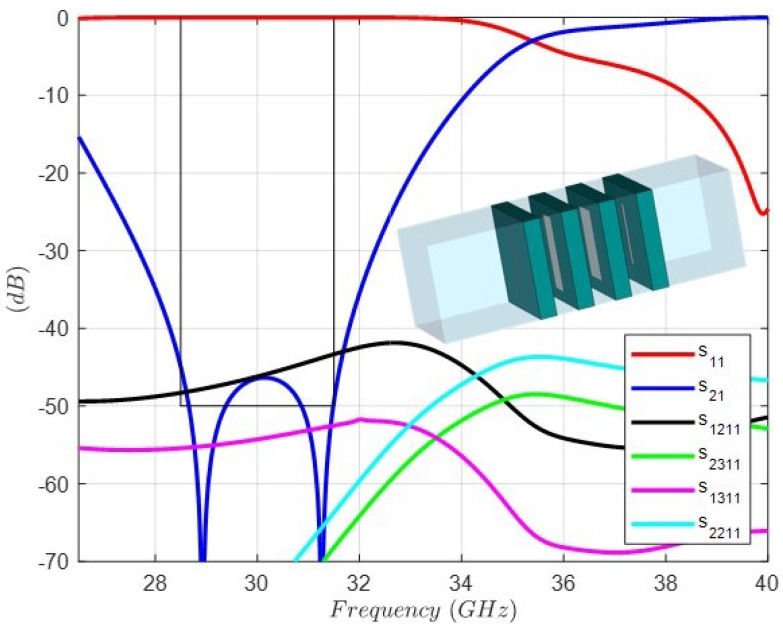
Full-wave response of the FSS unit cell topology and circuit values for the band-stop response with dielectric substrate $\epsilon_{r}=3.2$, thickness = 0.762 (mm). Dipole 1: $Length_{1}$ = 3.53 (mm), $Width_{1}$ = 0.13 (mm). Dipole 2: $Length_{2}$ = 3.43 (mm), $Width_{2}$ = 1.07 (mm). Dipole 3: $Length_{3}$ = 3.24 (mm), $Width_{3}$ = 1.44 (mm). Dipole 4: $Length_{4}$ = 2.83 (mm), $Width_{4}$ = 0.27 (mm). Period: 4.45 (mm). All lengths between substrates: 1 (mm).

**Figure 13 sensors-24-04452-f013:**
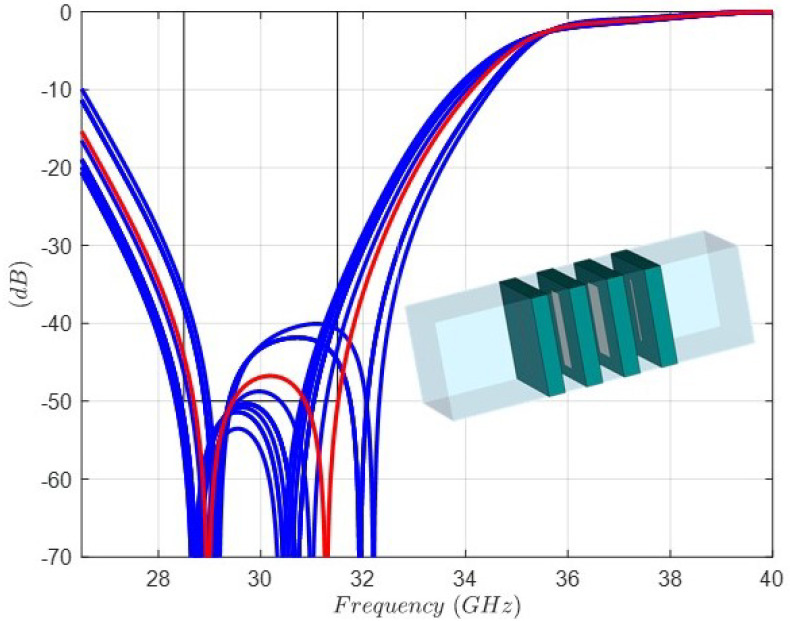
Sensitivity analysis considering a normal distribution with ±50 μm of variation in the lengths and widths of the four dipoles: full-wave response of $|s_{21}|$ parameter of the FSS unit cell with nominal values (red color response) in the caption of Figure 12.

**Figure 14 sensors-24-04452-f014:**
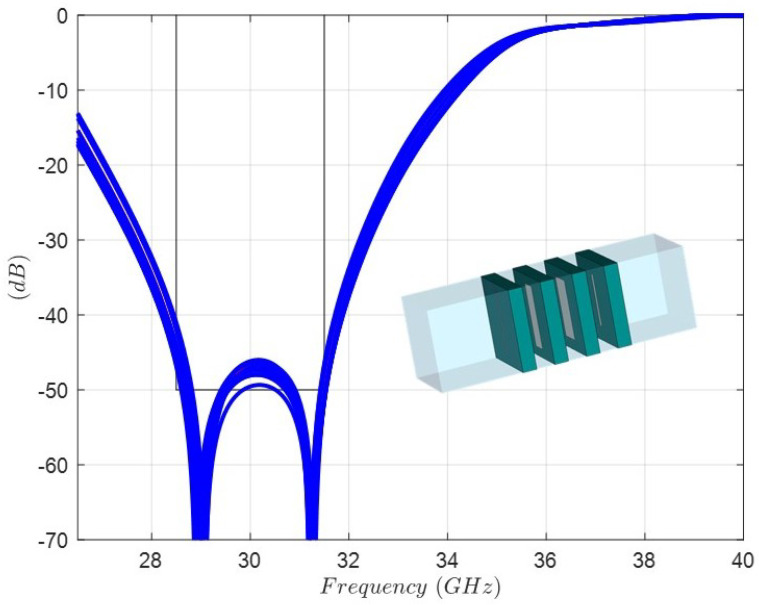
Sensitivity analysis considering a normal distribution with ±80 μm of variation in the thickness of the four Rohacell layers: full-wave response of $|s_{21}|$ parameter of the FSS unit cell.

**Figure 15 sensors-24-04452-f015:**
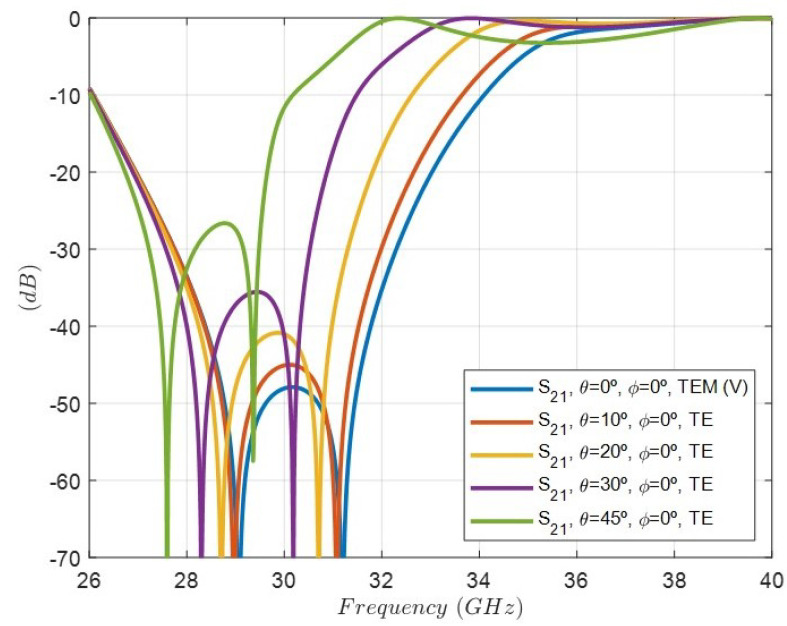
Simulated response of the rejection, i.e, the $|s_{21}|$ parameter of the FSS unit cell under oblique incidence for four different angles and TE excitations: (θ=10°, ϕ=0°) (θ=20°, ϕ=0°) (θ=30°, ϕ=0°) (θ=45°, ϕ=0°).

**Figure 16 sensors-24-04452-f016:**
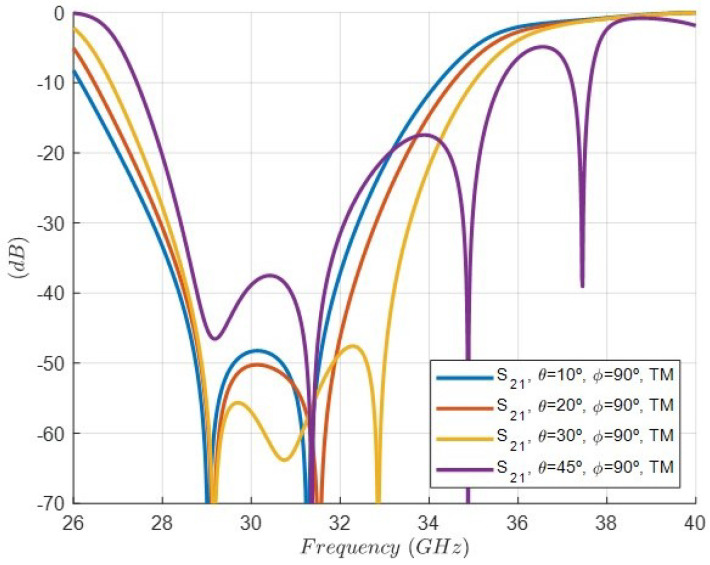
Simulated response of the rejection, i.e, the $|s_{21}|$ parameter of the FSS unit cell under oblique incidence for four different angles and TM excitations: (θ=10°, ϕ=90°) (θ=20°, ϕ=90°) (θ=30°, ϕ=90°) (θ=45°, ϕ=90°).

**Figure 17 sensors-24-04452-f017:**
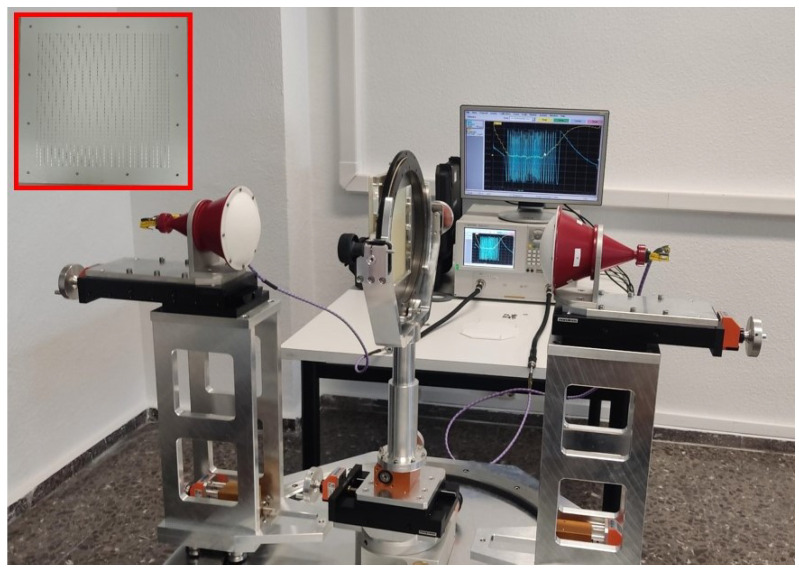
Measurement bench composed of the VNA and the radiation horns to obtain a plane wave impinging over the FSS. In the inset, the FSS is shown.

**Figure 18 sensors-24-04452-f018:**
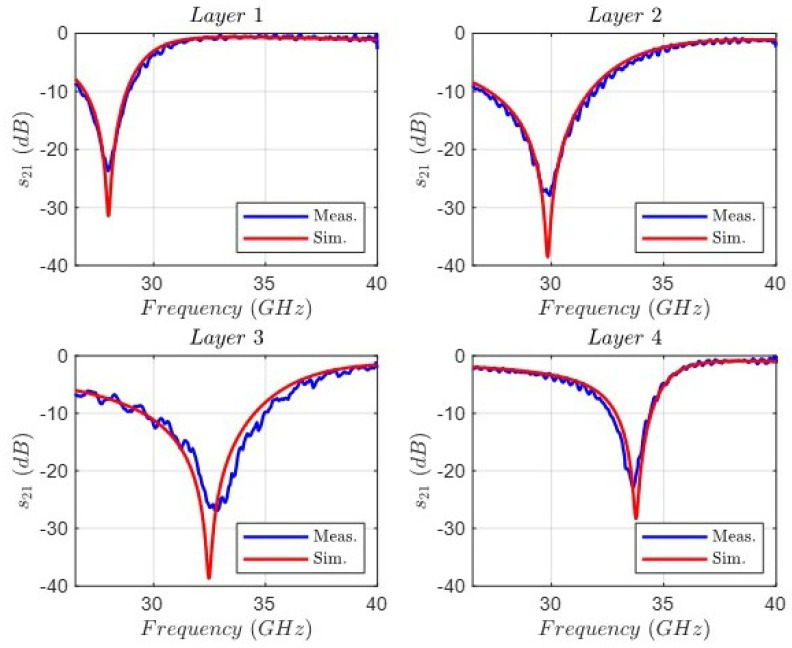
Comparison between the full-wave simulation and the measurement of the transmission coefficient $|s_{21}|$ for the four layers.

**Figure 19 sensors-24-04452-f019:**
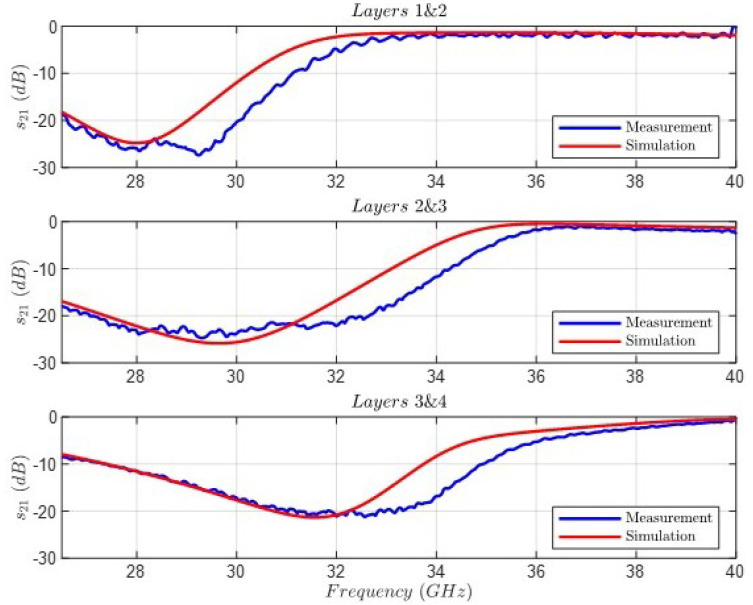
Comparison between the full-wave simulation and the measurement of the transmission coefficient $|s_{21}|$ for the three groups of two layers.

**Figure 20 sensors-24-04452-f020:**
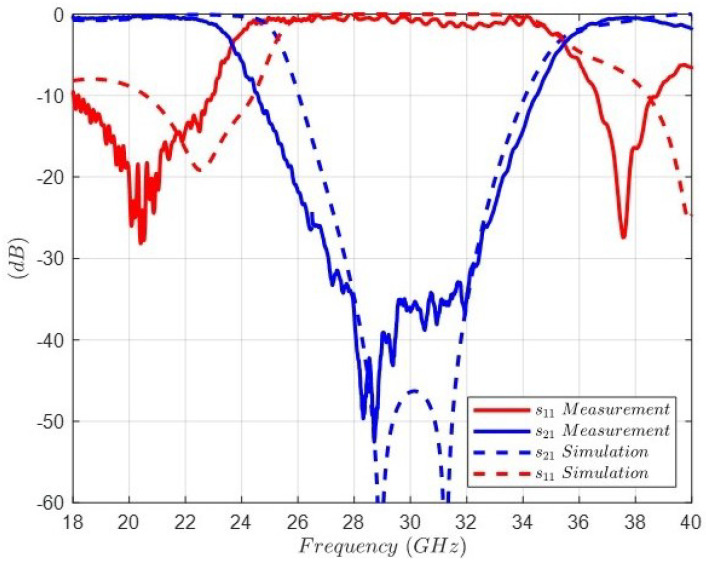
Comparison between the full-wave simulation and the measurement of the transmission coefficient $|s_{21}|$ and the reflection coefficient $|s_{11}|$ of the band-stop filter.

**Figure 21 sensors-24-04452-f021:**
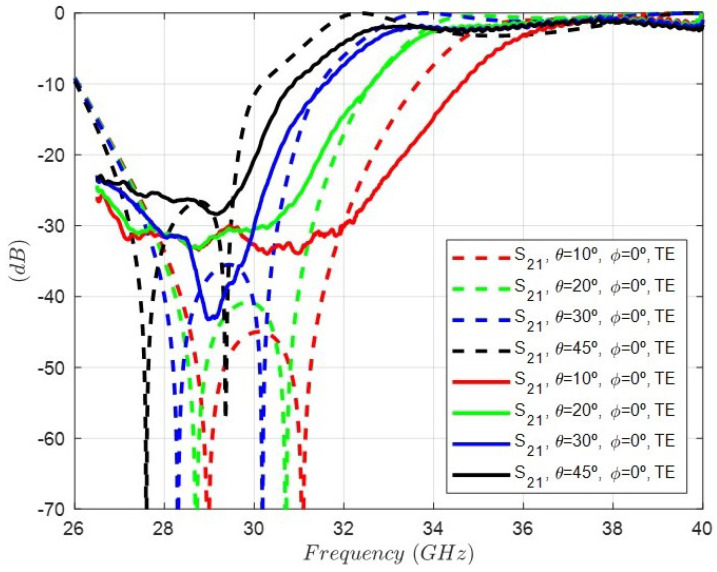
Comparison between the full-wave simulation (dashed line) and the measurement (solid line) of the transmission coefficient $|s_{21}|$ at oblique incidence for TE polarization and several angles.

**Figure 22 sensors-24-04452-f022:**
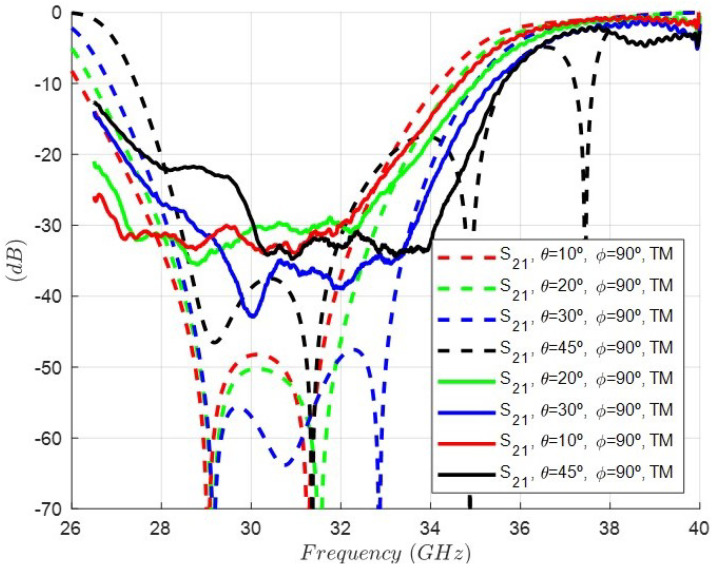
Comparison between the full-wave simulation (dashed line) and the measurement (solid line) of the transmission coefficient $|s_{21}|$ at oblique incidence for TM polarization and several angles.

**Table 1 sensors-24-04452-t001:** Band-stop filter specification.

Order N	4
Center Frequency $f_{o}$	30 GHz
Bandwidth BW	3 GHz
Attenuation	50 dB
Return Loss	18 dB

**Table 2 sensors-24-04452-t002:** Comparison of 2D band-stop frequency-selective surfaces’ (FSSs’) responses at normal incidence.

Reference	Theoretical Synthesis	Transmission Zeros Control	Attenuation Level Control	Attenuation (dB) Sim./Meas.	Center Frequency (GHz)	Bandwidth (%)	Filter Order
[6]—2010	NO	Heuristic	Heuristic	10/8	10.25	44	2
[20]—2016	NO	Heuristic	Heuristic	10/9	9.7	78	3
[17]—2017	NO	Heuristic	Heuristic	30/29	13.75	55	2
[11]—2018	NO	Heuristic	Heuristic	40/35	13	17	2
[9]—2020	NO	Heuristic	Heuristic	10/-	12	50	1
[28]—2022	NO	Heuristic	Heuristic	10/10	27.2	27.6	2
[29]—2022	NO	Heuristic	Heuristic	30/30	3.8	36	2
[30]—2024	NO	Heuristic	Heuristic	10/10	3.96	40	1
This work	YES	YES	YES	48/35	30	10	4

## Data Availability

The data are contained within the article.

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
