# Peer review of "Band-Stop Frequency-Selective Surface (FSS) with Elliptic Response Designed by the Extracted Pole Technique"

_sensors, 2024, doi:10.3390/s24144452_

Round 1

Reviewer 1 Report

Comments and Suggestions for Authors

1、  We have found many [???] in Section II, and there are significant discrepancies with the introduction mentioning "the theoretical synthesis of the band-stop filter using the Generalized Chebyshev Function and the extracted-pole technique."

2、  Is the illustration in Figure 2 reasonable?

3、  The paper mainly discusses the design and performance under normal incidence. For practical applications, changes in incidence angle and polarization state may affect the performance of the FSS. It is recommended to expand the response analysis under different incidence conditions and propose corresponding optimization design methods.

4、  The paper conducted a sensitivity analysis on manufacturing errors but only considered length and width variations of ±50μm. It is recommended to further study the impact of a wider range of manufacturing errors on performance and propose corresponding compensation design methods.

5、  The conclusion section should summarize the main findings and results of the study and explain their significance and impact. The content of your conclusion section is more suitable to be placed in the Experimental Results section. The conclusion section should summarize the main findings and results of the study and explain their significance and impact. It is recommended to rewrite the conclusion section.

6、  There are some words and grammar issues in the text that need to be addressed with a comprehensive revision.

Reviewer 2 Report

Comments and Suggestions for Authors

1. In the introduction, there are references to FFS. What does FFS stand for? Please check the writing conventions. In addition, what is elliptic response? It should be explained in the introduction.

2. How were the dimensions of the dipole obtained in Chapter 3 and how were the component values in Figure 4 determined?

3. The designed band-stop filter seems to lack practical significance as both its angular stability and polarization stability are not very good. Please explain the practical application scenarios and innovative points of this structure.

4. What is the reason for the difference in return loss between the solid and dashed lines in Fig. 5?

5. What is the reason for the relatively large discrepancy between the simulated and measured results outside the stopband in Fig. 19?

6. The references listed in Table 2 is relatively outdated. It is suggested to include relevant work from the past three years.

7. Can the response of the band-stop filter be extended to Nth order using the technology described in this paper?

8. The clarity of images such as Figure 17 and Figure 18 in the text is too poor. It is suggested to improve the quality of all images in the paper.

9. It is suggested to supplement test results under different incident angles and polarization.

10. The structure designed in this paper is symmetrical, but as the incident angle increases, there is a significant difference in the trend of TE and TM polarization in Figure 14 and Figure 15. Please explain this phenomenon.

Comments on the Quality of English Language

The quality of English needs to be improved, especially in terms of vocabulary writing errors.

Reviewer 3 Report

Comments and Suggestions for Authors

Your paper is good. Thank you!

Comments:

1. Please, change in text all "s21" to "magnitude |s21|".

2. Delete comma before "where" at line 177.

3. Delete one "the" at line 256.

Round 2

Reviewer 1 Report

Comments and Suggestions for Authors

The authors have addressed most of my concerns. However, I would suggest to reduce the size of the figures, some figures are too large.

Reviewer 2 Report

Comments and Suggestions for Authors

The author has fully answered all the questions raised.